# Isolation and Identification of Bacteria of Genus *Bacillus* from Composting Urban Solid Waste and Palm Forest in Northern Peru

**DOI:** 10.3390/microorganisms11030751

**Published:** 2023-03-15

**Authors:** Ernestina Vásquez, Carlos Millones

**Affiliations:** Biology Laboratory, Universidad Nacional Toribio Rodríguez de Mendoza de Amazonas, Chachapoyas 01001, Peru; ernestina.vasquez@untrm.edu.pe

**Keywords:** amylolytic activity, cellulolytic activity, DNA barcoding, urban solid waste

## Abstract

A technical challenge for composting in Peruvian cities with annual temperatures below 20 °C is that the degradation of municipal solid waste (MSW) is slow, so the identification of cold-adapted bacteria would be interesting for use as inoculants in places with these climatic conditions. This study isolated, identified, and evaluated bacterial strains with cellulolytic and amylolytic activities at low temperatures. Bacterial strains were isolated from the Chachapoyas Municipal Composting Plant and soil from the Ocol Palm Forest in northern Peru. The screening was carried out to evaluate the extracellular enzyme activity of the strains at low temperatures, grouping those with cellulolytic and cellulolytic/amylolytic activities. The DNA-barcoding using 16S rRNA and enzyme activity allowed the identification and selection of five species with enzymatic activity at 15 and 20 °C of the genus *Bacillus*, three with cellulolytic/amylolytic activity (*B. wiedmanii*, *B. subtilis*, and *B. velezensis*), and two with cellulolytic activity (*B. safensis* subsp. *safensis*, and *B. subtilis*). These strains showed tolerance to temperatures below optimum and could be used in further studies as inoculants for composting organic wastes at temperatures below 20 °C.

## 1. Introduction

The rapid population growth in many cities around the world has led to a considerable increase in municipal solid waste [1]. The municipal organic solid waste generated in markets and households is mainly composed of food waste, kitchen scraps (vegetables, peels, seeds, fruits), and yard waste, whose inadequate disposal in landfills causes the emission of greenhouse gases as they decompose, affecting the ecosystem [2,3]. Therefore, these wastes are mainly composed of cellulose and starch [4,5] and require the extracellular enzymatic action of cellulase and amylase for their degradation into simple molecules [6,7].

Cellulases are enzymes responsible for the degradation of cellulose, the main polymer present in the cell wall of plant cells that make up plant biomass. Cellulose chemically consists of several β-glucose subunits linked by β-1,4-glucosidic bonds. Cellulases break the bonds of this molecule [8,9]. The intermolecular bonds of cellulose create a particularly very tough crystalline molecule; therefore, the cellulolytic activity requires multiple complex enzyme systems for enzymatic hydrolysis, requiring the synergistic action of endo-glucanase, exo-glucanase, and glucosidase enzymes that degrade cellulose converting the products into a usable energy source [1]. Although the most efficiently active cellulases are of fungal origin, in recent years, there has been increased interest in cellulase produced by bacteria because they have a higher growth rate in short periods compared to fungi, giving them a competitive advantage during the process, highlighting the bacterial genera *Bacillus*, *Pseudomonas*, *Cellulomonas* [1,10].

The application of cellulase in composting allows for improving cellulose degradation and promoting the formation of humic substances allowing for obtaining a high-quality compost [11]. An alkaline cellulase produced by *B. licheniformes* 380 had the potential to accelerate the degradation of organic matter, degrading cellulosic residues in the initial stage of composting to optimize this process [9]. Cellulase enzyme activity and the use of microbial inoculum in composting vegetable waste accelerate the process and improve the quality of the final product [12]. Enzyme production is controlled by the microorganisms, and therefore factors such as pH, temperature, incubation period, and carbon and nitrogen sources should be evaluated to improve enzyme productivity [8].

Starch residues in kitchen waste can also be converted into simple sugars using amylase produced by microorganisms. Amylase degrades starch by breaking alpha-1,4-glycosidic bonds to produce linear and branched oligosaccharide chains. Compost bacteria belonging to the genus *Bacillus* can use starch as a substrate and catalyze the hydrolysis of α-D-(1, 4) glycosidic linkage [13,14].

Temperature is a critical factor for enzymatic reactions to take place. The speed of these reactions tends to decrease at low temperatures. However, there are microorganisms considered to be psychrotolerant, isolated from terrestrial environments, which are adapted to cold and can grow and develop at extreme temperature fluctuations [15]. The psychrophilic enzymes of these microorganisms must adapt to maintain adequate catalytic rates for cellular function [16].

Psychrophilic bacteria are those that, during the evolutionary process, have developed mechanisms of cold tolerance, having the capacity to grow and reproduce in low-temperature environments and capable of degrading organic macromolecules such as cellulose, hemicellulose, lignin, and starch for their utilization [17]. There are cold-adapted psychrotolerant bacterial strains that produce a higher number of extracellular enzymes, which play an important role in metabolic activities in the composting process at low temperatures [18]. The high level of low-temperature specific activity of cold-adapted enzymes is a key mechanism to compensate for the exponential decrease in chemical reaction rate as the temperature is reduced [17].

The composting process of urban solid waste in cold climate regions can be enhanced by adding psychrotolerant bacteria. In this regard, researchers isolated five bacterial strains of *Bacillus wiedmanii* from raw milk stored in a silo from a dairy processing plant, with minimal growth at or below 7 °C, with their optimal growth being between 20 and 40 °C [19]. Bacterial strains of the *Bacillaceae* family from the Chumbu Glacier, identifying *B. wiedmanii, B. velezensis, Paenibacillus odorifer*, and *Lysinibacillus fusiformis*, which recorded growth temperature range between 0 to 40 °C, with the optimum growth temperature being between 20 and 30 °C [20]. However, few investigations have been carried out to identify psychrotolerant bacterial strains with the potential for cellulose and amylose degradation for possible use in municipal solid waste composting at low temperatures.

In the study, bacterial strains were isolated from Chachapoyas Municipal Composting Plant (CMCP) and soil from the Ocol Palm Forest (OPF), characterized by their adequate capacity to degrade cellulose and starch at low temperatures. The main goal of this study was to isolate, identify, and evaluate bacterial strains with cellulolytic and amylolytic activities at low temperatures that could be used in further studies as inoculants for composting organic wastes at temperatures below 20 °C.

## 2. Materials and Methods

### 2.1. Site and Study Material

Two types of samples were collected in January 2021 in districts characterized by cold climates. The first type of sample was organic USW collected from the CMCP located in Tuctilla, Amazonas region, Peru (06°14′21.2″ S, 77° 50′47.4″ W and 2 311 m.a.s.l), with a mean annual temperature of 15.6 °C [21]. Compost samples were collected at different strata of the composting piles. Approximately 500 g of the composting pile was obtained at different stages of composting (Tc-1 and Tc-2 mesophilic, Tc-3 and Tc-4 thermophilic, and Tc-5 cooling and maturity). The second type of sample corresponded to 2.5 kg of soil from the OPF, located in the district of Molinopampa, Amazonas region, Peru, with minimum temperatures reaching 7 °C and maximum temperatures of 25 °C [22]. Approximately 500 g of soil were collected from five different points (OC-21-A: 06°15′58.9″ S, 77° 34′ 14.3″ W, 2261 m.a.s.l; OC-21-B: 06°15′5.4″ S, 77° 34′27.3″ W, 2281 m.a.s.l; OC-21-C: 06 °15′49.1″ S, 77° 34′33.7″ W, 2306 m.a.s.l; OC-21-D: 06°15′44.2″ S, 77° 34′41.4″ W, 2357 m.a.s.l; and OC-21-E: 06°15′43.1″ S; 77° 34′41.1″ W, 2347 m.a.s.l). The collected samples were placed in coded self-sealing polyethylene bags and transported at 4 °C in a styrofoam box with cooling gel for further processing. 

### 2.2. Determination of pH, Temperature, Electrical Conductivity, and Humidity

The determination of temperature, pH and electrical conductivity (EC) of compost or soils was recorded in situ. The temperature was measured with a waterproof digital HI 9350005 compost thermometer HANNA (Salaj, Romania). The pH was recorded using a HI 99121 soil pH multiparameter HANNA (Salaj, Romania). EC was recorded with a HI 993310 soil activity direct conductivity measurement kit HANNA (Salaj, Romania). Moisture content was determined by the gravimetric method using the formula: (1)MC%=W−w/w∗100
where (*MC*) is the moisture content, *W* is the initial weight of the sample, and *w* is the constant weight after removing the sample from the oven at 110 °C [6]. 

### 2.3. Cellulolytic Capacity of Isolates

Cellulolytic bacteria were isolated with the serial dilution method using 10 g of homogenized sample sieved on a No. 1 U.S. sieve. Standard Testing Sieve of 2.00 mm opening and 90 mL of 0.1 g/100 mL peptone water as a diluent were used [23]. Serial dilutions (10^−1^ to 10^−6^) were made from the initial suspension. From each dilution, an aliquot of 100 µL was spread on the surface of plates with 1% Carboxymethyl Cellulose (CMC) medium (10 g CMC, 2 g tryptone, 4 g KH_2_PO_4_, 4 g Na_2_HPO_4_, 0.2 g MgSO_4_·7H_2_O, 0.001 g CaCl_2_·2H_2_O, 0.004 g FeSO_4_·7H_2_O, 15 g agar, pH adjusted to 7, and made with 1 L deionized water). The plates were inoculated in duplicate and incubated at 30 °C for 24 to 48 h. The growing colonies were selected and subcultured by streaking using plates with 1% CMC medium in duplicate and incubated at the same conditions as above. 

Subsequently, cellulose degradation in all the isolates was evaluated using the Congo Red test for the assay β-D glucanase activities [24]. The Congo red test consisted of covering the surface of the plate cultures with 1 g/100 mL Congo red dye for 15 min. Subsequently, these plates were incubated 1 h at room temperature with 1 M NaCl solution. Finally, a 2 g/100 mL acetic acid solution was added to the plates for 10 s. to clarify the halo formed around the colony. In the cultures where the halo was observed, the diameter of the colony and the halo were measured. 

The inoculum of bacterial colonies with cellulolytic activity was subcultured and purified on 1% CMC medium plates until axenic cultures were obtained. Finally, they were aseptically transferred to tubes with Luria Bertoni (LB) Agar (1 g tryptone, 0.5 g yeast extract, 0.5 g NaCl, 15 g agar, pH adjusted to 7, and made with 1 L deionized water), and incubated at 30 °C for 24 h [23]. Pure cultures were coded according to sampling location, sample number, and random number according to colony isolation. A 20 mL/100 mL glycerol solution was added to the cultures, and they were kept frozen for later tests [25]. 

### 2.4. Amylolytic Capacity of Isolates 

Bacteria with cellulolytic capacity isolated at 30 °C were cultured in 2% Starch Agar (SA) medium (3 g meat extract, 6 g NaCl, 2 g starch, 12 g agar, pH adjusted to 7, and made with 1 L deionized water). After 24 to 48 h, the plates were covered with a lugol solution (2 g of potassium iodide and 1 g of iodine in 300 mL of water) to show the clear zones of degradation due to amylase activity [26]. 

### 2.5. Bacterial Enzyme Activity Screening at 15 °C

The cellulolytic and amylolytic activities of the isolates were preliminarily tested at low temperatures (i.e., 15 °C), for which the isolated strains were sub-cultured in triplicate by puncture in Petri dishes containing 1% CMC medium and 2% SA medium. The cultures were incubated at 15 °C in a refrigerated incubator using a FOC Cooled Incubator VELP. Scientific (Usmate, Italy), for 24 to 72 h. After the incubation period, bacterial enzymatic activity was screened using the Congo Red and the Lugol tests described above. The presence of transparent halos resulting from the hydrolysis of cellulose and starch was measured. The determination of the cellulolytic index (CI) or amylolytic index (AI) was calculated using the equation proposed by Naresh [27]: (2)$$\mathrm{CI}\;o\;\mathrm{AI}\;=\frac{\left(Halo\;diameter-Bacterial\;colony\;diameter\right)}{Bacterial\;colony\;diameter}$$

### 2.6. Phenotypic Identification of Bacterial Isolates

By using conventional bacteriological methods, a total of 20 isolated bacterial strains were characterized based on morphological, biochemical, and physiological features, as well as the characteristics of the colony in a solid medium. The morphological characterization of the bacteria was performed by Gram staining using a DM500 binocular light microscope Leica (Heerbrugg, Switzerland). The diameter and length of the bacteria were determined using a BX53 professional microscope Olympus (Tokyo, Japan). The biochemical and physiological evaluations were according to the Bergey Manual of Systematic Bacteriology [28]. Briefly, the performed tests included catalase, indole, citrate, gelatin hydrolysis, and hemolysin productions. Finally, the development in 1% CMC broth was evaluated at different temperatures (10, 20, 30, 40, 50, and 60 °C). The tolerance to NaCl at different concentrations (5, 10, and 20%) and pH levels (5, 6, 7, 8, 9, and 10) were also evaluated [25]. The characteristics of the culture and appearance of the colony (size, color, shape, edge, appearance, and smell) were described according to standard methods.

### 2.7. DNA Sequencing and Phylogenetic Analysis of Bacterial Isolates

Bacterial isolates with cellulolytic and amylolytic activities at 15 °C were identified using three molecular markers (i.e., 16S ribosomal RNA; *gyr*A, DNA gyrase subunit A; and *rpo*B, RNA polymerase β subunit). Briefly, isolates were superficially scraped from Petri dishes with a sterilized scalpel and placed in prelabeled 2.0 mL Safelock Eppendorf tubes. Genomic DNA was extracted using a DNA Miniprep Kit (ZymoBIOMICS, California, USA) following the manufacturer’s instructions. The primer combination for the three markers was as follows: for 16S, GM3F (5′-AGAGTTTGATCMTGGC-3′) and GM4R (5′-TACCTTGTTACGACTT-3′) [29]; for *gyr*A, 42f (5′-CAGTCAGGAAATGCGTACGTCCTT-3′) and 1066 (5′-CAAGGTAATGCTCCAGGCATTGCT-3′) [30]; and for rpoB, 2292f (5′-GACGTGGGATGGCTACAACT-3′) and 3354r (5′-ATTGTCGCCTTTAACGATGG-3′) [31]. Each marker was amplified using polymerase chain reaction (PCR) with MasterMix (Promega, Madison, WI, USA) in the following reaction mixture: 10 ng of DNA and 0.25–0.5 pmol of the forward and reverse primers in a total volume of 10 μL. The PCR protocol for the three markers followed a predenaturation step (95 °C for 3 min), 35 cycles of denaturation (94 °C for 1 min), annealing (55 °C for 1 min), and extension (72 °C for 1 min); and a final elongation step (72 °C for 5 min). Amplicons were purified using a Macherey-Nagel Kit (NucleoSpin Gel and PCR Clean-up, Düren, Germany). Sequences of the forward and reverse strands were commercially determined by Macrogen (Seoul, South Korea) and then edited in Chromas v1.45 software [32]. A total of 34 sequences were generated and deposited in GenBank (Appendix A).

Sequences were initially aligned using MUSCLE algorithms [33] and manually adjusted with MEGA7 [34]. The nucleotide substitution model that best matched each marker was selected using PartitionFinder [35]. The best partitioning strategy and the best sequence evolution model were selected based on the Bayesian information criterion (BIC). Maximum likelihood (ML) analysis was performed with raxmlHPC-AVX implemented in the raxmlGUI 1.3.1 interface [36] using the best model as appropriate with 1000 bootstrap replicates for both datasets. Intraspecific and interspecific pairwise divergence was estimated using the p-distance model in MEGA7.

### 2.8. Bacterial Extracellular Enzyme Activities by Reducing Sugars

In assessments of bacterial extracellular enzyme activities by reducing sugars, the 3,5-dinitrosalicylic acid (DNS) method was used [37]. This method measures the accumulation of products after hydrolysis, such as reducing sugars. The strains were cultivated in test tubes using 10 mL of 2% starch broth and 1% CMC broth and incubated at 10, 15, and 20 °C for 72 h. Cultures were centrifuged at 4000 rpm using an MPW-251 centrifuge (Warszawa, Poland) for 15 min, and the supernatant was used as an enzyme solution. 0.1 mL of the enzyme was added to 1 mL of a 1 g/100 mL starch buffer solution at pH 7. This mixture was incubated for 10 min at 37 °C. In order to stop the enzymatic reaction, 2 mL of DNS reagent was added (1 g dinitro salicylic acid, 16 g NaOH, 300 g sodium potassium tartrate, and distilled water, up to 1 L). Then it was boiled for 10 min for staining and then cooled in an ice water bath for 2 min to stabilize the staining. Subsequently, 200 μL were taken and diluted with 2.5 mL of distilled water to measure their optical density at 540 nm using a UviLine 9400 spectrophotometer (Shanghai, China). The blank (negative control) contained all the reagents except for the enzyme. The glucose concentration was determined based on the standard curve with the absorbance readings. Enzyme activity was determined using a calibration curve for -D-glucose [27]. One unit (U) of enzyme activity was determined as the amount of enzyme required to release 1 μmole of reducing sugar per minute under assay conditions. 

### 2.9. Analysis of Data

A cluster analysis was performed based on squared Euclidean distances to form groups of bacterial strains with metabolic efficiency at low temperatures (15 °C) using the cellulolytic index and amylolytic index data. Data analysis was performed with the SAS (Statistical Analysis System) for the Windows V8 package. To analyze the metabolic activities of the strains with higher cellulolytic and amylolytic activities at low temperatures, cellulolytic and amylolytic indexes and reducing sugars in CMC and starch broth were analyzed through multivariate analysis of variance where Factor A is the bacterial strains, and Factor B is growth temperature (15 °C and 20 °C). Significant differences were analyzed at ρ ≤ 0.05 and ρ ≤ 0.01, respectively. Normality was previously checked with the Shapiro-Wilk test and the homogeneity of the variances. The design was balanced with all the samples analyzed under similar conditions, and there were no missing data. Differences between the different types of metabolic activities (treatments) were evaluated using analysis of variance (ANOVA) of two factors and post-hoc Tukey HSD. The CI and AI data were square-root-transformed (X + 0.5)^1/2^. The relationship between the metabolic activities was performed using the principal component analysis (PCA) using the package ggbiplot. Principal component results were visualized with the Biplot built between the first two principal components (PC1 and PC2). All statistical analyzes were conducted in R software version 4.1.0.

## 3. Results

### 3.1. Physicochemical Characterization of the Collected Samples

The physicochemical characterization of the compost samples collected from CMCP in Tuctilla showed differences in temperature, pH, EC, and humidity in the mesophilic, thermophilic, cooling, and maturation stage. Meanwhile, the soil samples collected from the Ocol Palm Forest showed differences in temperature, pH, EC, and humidity Oc-A, Oc-B, Oc-C, Oc-D, and Oc-E (Table 1).

### 3.2. Bacterial Isolation and Screening for Cellulolytic and Amylolytic Activities at 30 °C and 15 °C

Sixty bacterial strains were isolated from USW at different stages of degradation from CMCP using 1% CMC medium and 2% SA medium. In these media, 55 strains showed cellulolytic/amylolytic activities at 30 °C, while five strains showed only cellulolytic activity. Additionally, 17 strains from the Tc-1 pile and nine strains from the Tc-2 pile were in the mesophilic stage; eight strains from the Tc-3 pile and 12 strains from the Tc-4 pile were in the thermophilic stage, and 14 strains from the Tc-5 pile were in the cooling and maturation stage. The screening in CMC medium and SA medium at low temperature (15 °C) reported only 49 strains, of which 39 had cellulolytic/amylolytic activities, and 10 strains showed cellulolytic activity solely. From these, 16 strains from the Tc-1 pile and eight strains from the Tc-2 pile were in the mesophilic stage; 7 strains from the Tc-3 pile and 12 strains from the Tc-4 pile were in the thermophilic stage, and six strains from the Tc-5 pile were in the cooling and maturation stage. 

Twelve bacterial strains were isolated from OPF at different types of soils using 1% CMC medium and 2% SA medium. In these media, seven strains showed cellulolytic/amylolytic activities at 30 °C. The screening in CMC medium and SA medium at low temperature (15 °C) reported only seven strains, of which two had cellulolytic/amylolytic activities, and five strains showed amylolytic activity solely. From these, five strains of amylolytic from Oc-A (1), Oc-B (1), and Oc-E (3), and two strains of cellulolytic/amylolytic from Oc-A and Oc-E. 

### 3.3. Bacterial Extracellular Enzyme Activity at 15 °C

Bacterial strains with cellulolytic or amylolytic activity were grouped by means of the cluster analysis obtained by Ward’s method (Figure 1). In the CMCP samples, 49 isolates showed adequate development at 15 °C (Figure 1a,b), and 14 strains showed high levels of cellulolytic or amylolytic activity, including the isolates Tc-1-16 to Tc-4-62 (Figure 1a). The diameter of the cellulose hydrolysis halos ranged from 7.50 to 8.6 mm (Figure 1e), and the CI varied from 0.33 to 0.9. Conversely, eight strains showed better amylolytic activity at 15°C, including the strains Tc-2-25 to Tc-5-82 (Figure 1b). The diameter of the starch hydrolysis halo ranged from 10.6 to 12.8 mm (Figure 1f), and the AI varied from 2.01 to 2.57. The strains with the highest cellulolytic or amylolytic activity at 15°C were Tc 2-30, Tc 2-29, Tc 5-81, and Tc 5-82 (Figure 1a,b). 

The cluster analysis for the cellulolytic bacterial activity at 15 °C from the OPF revealed activity in only two strains (i.e., Oc-A-10 and Oc-E-31). These strains developed halos for cellulose hydrolysis measuring 2.60 to 5.30 mm and showed CI ranging from 1.6 to 2.53 (Figure 1c). The cluster analysis for the amylolytic activity from the OPF revealed that three of the seven strains showed larger hydrolysis halos (diameter 6.9 to 8.5 mm) and higher AI ranging from 1.76 to 5.85 (Figure 1d).

### 3.4. Molecular Identification and Phylogenetic Analysis of Bacterial Isolates

Twenty bacterial isolates from CMCP composting piles and OPF soils with the highest extracellular enzyme activities were molecularly identified using three markers (i.e., 16S rRNA, *gyr*A, and *rpo*B). These isolates clustered in the following five species *B. cereus*, *B. safensis* subsp. *safensis*, *B. subtilis*, *B. velezensis*, and *B. wiedmannii* (Figure 2).

### 3.5. Phenotypic Characterization of Bacterial Isolates

#### 3.5.1. Morphological and Biochemical Analyses

In the morphological characterization, all the strains were Gram-positive bacilli grouped mostly in chains, with oval edges and visible spore formation in the Gram stain. Dimensions are shown in Appendix A. In the biochemical characterization, all the strains were aerobic bacilli being catalase enzyme and hemolysin producers. None of the strains tested positive for indole and citrate utilization. Regarding the production of gelatinase, they were negative except for the strains Oc-A-10, Tc-3-42, Tc-1-16, and Tc-4-62 (Appendix A).

#### 3.5.2. Physiological Characterization

Table 2 shows the physiological behavior of strains at different temperatures. Most of the strains did not grow at 10 °C, except for the strains Tc-2-29, Tc-3-47, Oc-E-31, Tc-4-67, and Tc-5-81, which slightly grew. All the strains showed properly developed at 20, 30, and 40 °C. Strikingly, strains Tc-4-68, Oc-A-10, Oc-E-31, Tc-4-66, Tc-3-42, Tc-2-33, Tc-4-67, Tc-1-16, and Tc-4-62 showed good growth at 50 °C. No bacterial strain developed at 60 °C. Regarding the NaCl tolerance test, all the tested strains only tolerated the 5% concentration, but not higher concentrations. In relation to the tolerance at pH variation, all the strains showed optimal growth at pH 6, 7, 8, and 9.

### 3.6. Bacterial Extracellular Enzyme Activity

Based on the results of the microbiological and molecular characterization, as well as the extracellular enzyme activity (i.e., CI, AI, and polysaccharides hydrolysis), seven strains were used for the determination of cellulase and amylase activities, four of which were *B. wiedmanii* (Tc-2-25, Tc-2-28, Tc-2-29, Tc-5-82), and the other three were *B. subtilis* (Oc-A-10), *B. velezensis* (Oc-E-31), and *B. subtilis* (Tc-3-42).

The bifactorial MANOVA of the variables CI, AI, and cellulases and amylases activities of strains isolated from CMCP and OPF revealed that the following factors strains, temperatures, and also interaction effect of strains and temperature were highly significant (ρ ≤ 0.01) on the bacterial metabolism of the isolated strains (Table 3). 

The extracellular enzyme activity evaluated through the CI and AI of the bacteria isolated from CMCP and OPF was different at the two temperatures (i.e., 15 and 20 °C). Under a temperature of 10 °C, there was no growth of bacterial strains studied. The bacterial strains *B. wiedmanii* (Tc-2-29) and *B. subtilis* (Tc-4-42, Oc-A-10) cultivated in 1% CMC medium registered a similar CI at both temperatures. However, the latter strains showed higher metabolic activity (ρ ≤ 0.05) (Figure 3a). It suggests the tolerance of these strains to 15 and 20 °C. Meanwhile, the bacterial strains *B. wiedmanii* (Tc-2-25, Tc-2-28, Tc-2-29, and Tc-5-82), as well as *B. subtilis* (Tc-3-42 and Oc-A-10), cultured in SA medium showed a similar AI at 15 and 20 °C (Figure 3b), showed tolerance in their growth and metabolic activity. Additionally, all strains show greater metabolic effectiveness in the SA medium compared to the CMC medium based on the higher values of the AI (Figure 3a,b). 

The enzyme activity through the degradation of polysaccharides to reducing sugars by extracellular enzymes of the isolated strains from CMCP and OPF showed differences at 15 and 20 °C. Thus, the bacterial strains *B. wiedmanii* (Tc-2-25, Tc-2-28, Tc-2-29, Tc-5-82), *B. subtilis* (Tc-3-42), and *B. velezensis* (Oc-E-31) cultured in 1% CMC broth showed a similar degradation of cellulose when reducing sugars at both temperatures; however, the *B. subtilis* strain showed greater degradation activity over the other two strains (ρ ≤ 0.05) (Figure 3c). It confirms tolerance during growth and extracellular enzyme activity at low temperatures. Meanwhile, the bacterial strains *B. wiedmanii* (Tc-2-28) and *B. subtilis* (Oc-A-10) cultivated in starch broth at 2% registered similar degradation of starch when reducing sugars at both temperatures; however, the strains *B. subtilis* (Tc-3-42, Oc-A-10) and *B. velezensis* (Oc-E-31) showed higher extracellular enzyme activity at 15 °C (ρ ≤ 0.05) (Figure 3d). Additionally, all strains had greater metabolic effectiveness when reducing sugars in 1% CMC broth compared to starch broth (Figure 3c,d).

The extracellular enzyme activities of the bacterial strains with cellulolytic and amylolytic activity subjected to two growth temperatures were evaluated through the analysis of principal components (PC), PC1, and PC2, which explain 97.67% of the variability of the data. (Figure 4). Positive correlations were registered in the CI and AI variables, lacking dependence between them. Therefore, the results show that the cellulolytic activity is different from the amylolytic activity from the evaluated strains and that they can be selected according to the type of extracellular enzyme activity.

## 4. Discussion

### 4.1. Cellulolytic and Amylolytic Activity

From the samples analyzed in five composting piles at different stages from CMCP, 60 bacterial strains were isolated. Likewise, 12 bacterial strains were isolated from OPF soil samples. In the selection of the strains with the highest metabolic activities (cellulolytic or amylolytic) at low temperatures, the cluster analysis included the diameter of the hydrolysis halo and the CI and AI. Hydrolysis halos are measures of the catalytic capacity of a microorganism due to the synthesis of enzymes such as cellulases or amylases [27,38]. The use of hydrolysis indices or radii allows a relationship between the diameter of the colony and the diameter of the hydrolysis halo [39]. This provides more precise values to qualitatively evaluate the bacteria metabolism. Hydrolysis indices have been confirmed when evaluating the metabolic activity of *B. amyloliquefaciens* after using starch hydrolysis ratios [40]. Additionally, CI has been used in *B. cereus* [41] and thermophilic bacteria [27]. AI was also used in *B. cereus* [42]. In this study, the use of hydrolysis halos, CI, and AI allowed the assessment of 20 bacterial strains with cellulolytic or amylolytic potential at 15 °C. 

### 4.2. Molecular Identification and Phylogenetic Analysis of Bacterial Isolates

The morphological simplicity of bacteria limited the correct identification of species from the CMCP piles and OPF soils. Our molecular approach using three molecular markers (i.e., 16S rRNA, *gyr*A, and *rpo*B) allowed the identification of five bacterial species with extracellular enzyme activity at low temperatures, namely *B. cereus* and *B. safensis* subsp. *safensis*, *B. subtilis*, *B. velezensis* and *B. wiedmanii*. The distinction among these *Bacillus* species could not be possible without a molecular characterization since the species of the genus *Bacillus* have similar morphological and biochemical features [20]. In this study, two strains of *B. subtilis* showed extracellular enzyme efficiency in the degradation of cellulose and starch. Thus, the strains Tc-1-16 and Tc-4-62 showed only cellulolytic activity. The *B. velezensis* is gram-positive with niches in the soil, water, roots, and fermented wood [43], as well as cold areas such as glaciers [20]. The strain of *B. velezensis* was isolated from the soils of the OPF, an area characterized by a humid and temperate climate. This strain registered good extracellular enzyme activity in the degradation of cellulose and starch to reducing sugars, especially at 15 °C (Figure 3d). The *B. cereus* group includes many species phylogenetically organized into three clades, where *B. anthracis* and *B. wiedmanii* are placed in Clade 1, group II [44,45]. *B. wiedmanii* strains have been confirmed to have tolerance at low temperatures [20,46]. Although 11 strains of *B. wiedmanii* were isolated from the CMCP, the strain TC-2-29 had the highest extracellular enzyme activity in the degradation of cellulose to reducing sugars at low temperatures (Figure 3c). *B. safensis* is composed of two subspecies (*B. safensis* subsp. *safensis* and *B. safensis* subsp. *osmophilus*) [47,48]. The strain *B. safensis* subsp. *safensis* (Tc-4-67) registered cellulolytic activity but lack of amylolytic activity [47]. 

### 4.3. Phenotypic Identification

Bacterial strains that hydrolyze cellulose or starch were also identified based on their morphology and biochemical characteristics. In the microscopic evaluation, *B. wiedmanii* recorded gram-positive bacilli and ellipsoidal endospores with a central position, 2.8 μm of average length, 1 μm of average width, and growth at 10–40 °C. These features have also been reported by Miller [19]. Additionally, although our eleven *B. wiedmanii* isolates were gelatinase-negative (Appendix A), Miller [19] reported 11 gelatinase-positive strains of *B. wiedmanii,* and six of these used citrate as a carbon source. It suggests the difficulty of phenotypic, biochemical, and even 16rRNA gene analyses to resolve differences in the systematics of the genus *Bacillus* [29]. Our study showed *B. safensis* subsp. *safensis* being gram-positive, spore-forming bacilli, and growing at 10–50 °C. Although these features are similar to those reported by Abril [47], other features in the bacillus, such as length (1.8 μm), width (0.5 μm), and negative gelatinase hydrolysis, were different from our strain. Furthermore, *B. subtilis* was gram-positive, spore-forming bacilli, 2.24 μm in average length, 0.56 μm in average width, growing at 20–50 °C, and gelatinase positive. Recently, Dunlap [49] found similar features in strains of *B. subtilis*. 

### 4.4. Bacterial Extracellular Enzyme Activity

In this study, the cellulolytic potential (extracellular cellulase production) of isolated bacteria was evaluated by the formation of a hydrolysis halo (~8.6 mm in diameter) in a CMC medium. This is corroborated by reports on *Bacillus* species being significant cellulase production [38,50,51,52]. The strains of *B. subtilis* (Oc-A-10, Tc-3-42, Tc-1-16, Tc-4-62) have cellulolytic activity due to the synthesis of endo and exocellulases. This has been previously reported in the composting of coffee residues [36] and cow manure [50]. The integrative approach (i.e., molecular identification, extracellular enzyme activities) performed in this study allowed the identification of bacterial strains psychrotolerant for their potential to future composting USW at low temperatures. Accordingly, a first consortium showing the highest amylolytic and cellulolytic activity could be composed of *B. subtilis*, *B. velezensis*, and *B. wiedmanii,* and a second consortium showing higher cellulolytic activity could be composed of *B. safensis* subsp. *safensis* and *B. subtilis*.

## 5. Conclusions

In this study, bacteria with cellulolytic and amylolytic activity, tolerant to low temperatures, were isolated and identified from organic wastes in different composting stages from the Chachapoyas Municipality Composting Plant and from soil from the Ocol Palm Forest, Amazonas, Peru. Five species stood out for their enzymatic activity at 15 and 20 °C; three with cellulolytic/amylolytic activity (*B. subtilis*, *B. velezensis*, and *B. wiedmanii*), and two with cellulolytic activity (*B. safensis* subsp*. safensis* and *B. subtilis*). The selected bacteria, due to their tolerance to temperatures lower than their optimum development, could be used in further studies as inoculants for organic composting waste at temperatures below 20 °C.

## Figures and Tables

**Figure 1 microorganisms-11-00751-f001:**
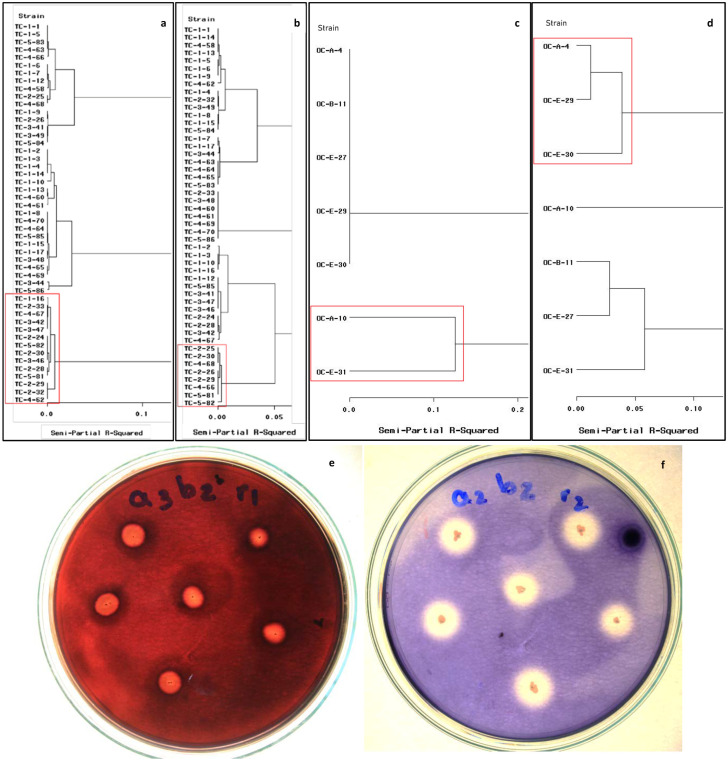
Cluster analysis (Ward’s method) for the bacterial extracellular enzyme activity at 15 °C. Cellulolytic activity (**a**) and amylolytic activity (**b**) from CMCP. Cellulolytic activity (**c**) and amylolytic activity (**d**) from OPF isolates. Strains showing high metabolic activities are a highlight in a red frame. Halos around bacterial colonies are indicative of cellulose degradation (**e**) and amylose degradation (**f**).

**Figure 2 microorganisms-11-00751-f002:**
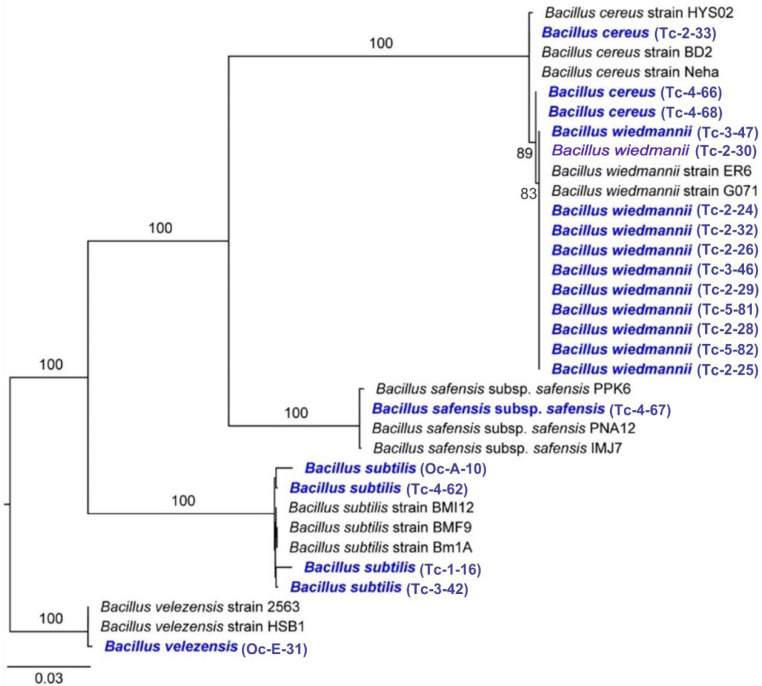
Maximum likelihood phylogram of bacterial isolates from CMCP and OPF based on three markers. Numbers along branches are RaxML bootstrap supports based on 1000 replicates. The legend below represents the scale for nucleotide substitutions.

**Figure 3 microorganisms-11-00751-f003:**
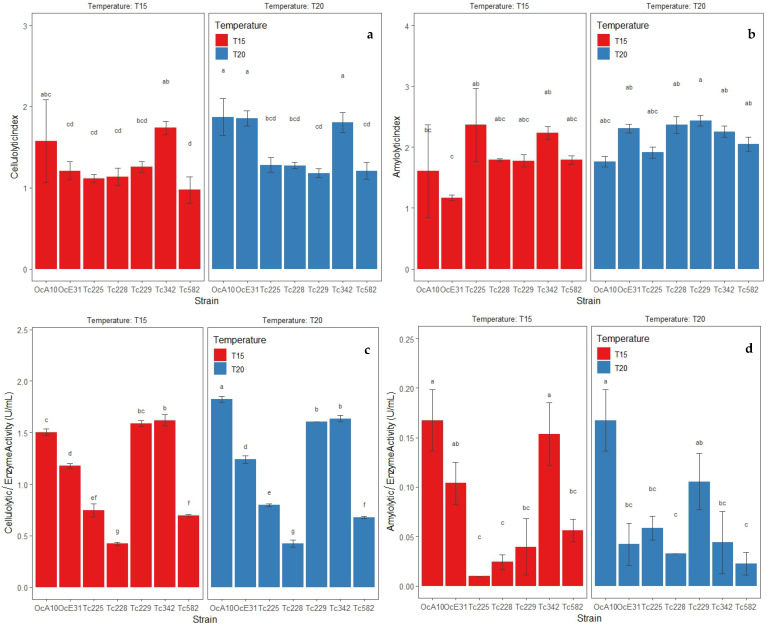
Metabolic activities by extracellular enzymes of bacterial strains isolated from CMCP and OPF at 15 and 20 °C. (**a**) Cellulolytic index (CI) of bacterial strains cultivated in 1% CMC medium, (**b**) amylolytic index (AI) of bacterial strains cultivated in SA medium, (**c**) enzyme activity in bacterial strains cultivated in 1% CMC broth, (**d**) enzyme activity in bacterial strains cultivated in starch broth. Data presented with means ± standard deviation. Different letters indicate significant differences in the parameters (ρ ≤ 0.05) according to Tukey’s Test.

**Figure 4 microorganisms-11-00751-f004:**
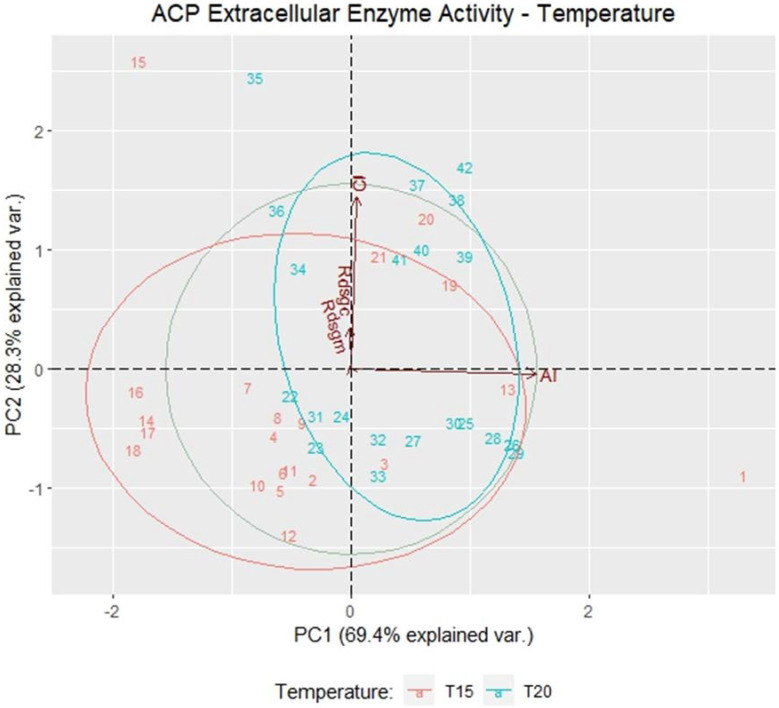
The main components of the extracellular enzyme activity of bacterial strains grown at 15 and 20 °C in the variables cellulolytic index (CI), amylolytic index (AI), enzyme activity in bacteria with higher cellulolytic activity (Rdsgc), and enzyme activity in bacteria with higher amylolytic activity (Rdsgm).

**Table 1 microorganisms-11-00751-t001:** Physicochemical characterization of the collected samples.

Stage/Code	Temp. °C	pH	EC mS·cm^−1^	Humidity %
**Compost samples from CMCP in Tuctilla**				
Mesophilic	29.5	4.07	1.57	72.4
Thermophilic	53.5	6.83	1.02	54.8
Cooling and maturation	36.0	7.04	1.07	38.0
**Soil samples Ocol Palm Forest**				
Oc-A	16.5	8.13	0.59	19.5
Oc-B	17.0	7.76	0.40	42.2
Oc-C	17.2	7.87	0.46	25.2
Oc-D	17.6	5.49	0.84	65.9
Oc-E	17.7	5.76	0.71	32.3

**Table 2 microorganisms-11-00751-t002:** Physiological characterization of the isolated strains regarding tolerance tests at different temperatures, pH, and NaCl concentrations.

Strain N°	Code	Specie	T (°C)						pH						NaCl (%)		
			10	20	30	40	50	60	5	6	7	8	9	10	5	10	20
1	Tc-2-25	*B. wiedmanii*	-	+	+	+	-	-	+	+	+	+	+	+	+	-	-
2	Tc-2-28	*B. wiedmanii*	-	+	+	+	-	-	+	+	+	+	+	+	+	-	-
3	Tc-2-29	*B. wiedmanii*	+	+	+	+	-	-	+	+	+	+	+	+	+	-	-
4	Tc-3-47	*B. wiedmanii*	+	+	+	+	-	-	+	+	+	+	+	+	+	-	-
5	Tc-4-68	*B. cereus*	-	+	+	+	+	-	+	+	+	+	+	+	+	-	-
6	Tc-5-82	*B. wiedmanii*	-	+	+	+	-	-	+	+	+	+	+	+	+	-	-
7	Oc-A-10	*B. subtilis*	-	+	+	+	+	-	+	+	+	+	+	+	+	-	-
8	Oc-E-31	*B. velezensis*	+	+	+	+	+	-	+	+	+	+	+	+	+	-	-
9	Tc-2-24	*B. wiedmanii*	-	+	+	+	-	-	+	+	+	+	+	+	+	-	-
10	Tc-2-30	*B. wiedmanii*	-	+	+	+	-	-	+	+	+	+	+	+	+	-	-
11	Tc-2-26	*B. wiedmanii*	-	+	+	+	-	-	+	+	+	+	+	+	+	-	-
12	Tc-4-66	*B. cereus*	-	+	+	+	+	-	+	+	+	+	+	+	+	-	-
13	Tc-3-42	*B. subtilis*	-	+	+	+	+	-	+	+	+	+	+	+	+	-	-
14	Tc-2-32	*B. wiedmanii*	-	+	+	+	-	-	+	+	+	+	+	+	+	-	-
15	Tc-2-33	*B. cereus*	-	+	+	+	+	-	+	+	+	+	+	+	+	-	-
16	Tc-4-67	*B. safensis* subsp*. safensis*	+	+	+	+	+	-	+	+	+	+	+	+	+	-	-
17	Tc-1-16	*B. subtilis*	-	+	+	+	+	-	+	+	+	+	+	+	+	-	-
18	Tc-3-46	*B. wiedmanii*	-	+	+	+	-	-	+	+	+	+	+	+	+	-	-
19	Tc-4-62	*B. subtilis*	-	+	+	+	+	-	+	+	+	+	+	+	+	-	-
20	Tc-5-81	*B. wiedmanii*	+	+	+	+	-	-	+	+	+	+	+	+	+	-	-

Note: “+” in the table represents growth, and “-” means no growth.

**Table 3 microorganisms-11-00751-t003:** Bifactorial MANOVA of the effects of strains and low temperatures and their combined effects on bacterial extracellular enzyme activity.

Factors	Wilk’s Lambda	F	p
Strains	0.0002	40.17	**
Temperature	0.2099	23.53	**
Strains x temperature	0.0190	7.79	**

** Significant difference to ρ ≤ 0.01.

## Data Availability

Appendix A, Morphological and biochemical features of the isolated bacterial strains in Appendix A, and Bacterial screening for amylolytic and cellulolytic activities in Appendix A.

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
