# Peer review of "Isolation and Identification of Bacteria of Genus Bacillus from Composting Urban Solid Waste and Palm Forest in Northern Peru"

_microorganisms, 2023, doi:10.3390/microorganisms11030751_

Round 1

Reviewer 1 Report

The stated aim of the research was to isolate cellulolytic and/or amylolytic bacteria from compost or from soil in the Amazonas region, Peru.  Specifically, the authors sought bacteria which may have utility in composting in regions with cold climates.  The manuscript describes obtaining Gram-positive bacterial strains that were selected at 30oC for their ability to degrade cellulose and these strains were then screened at 30oC for their ability to degrade amylose.  Subsequently, candidates showing potential for these enzymatic activities were tested for their capacity to make these enzymes at 15oC and 20oC, having established the initial enzymatic capacity at 30oC.  The isolates were classified into presumptive species based on sequence analysis of three highly conserved genes, which have been used previously to differentiate Bacillus species (at least to some extent).  Some phenotypic (limited number of biochemical tests) and physiological traits (size of rods) are also reported.  The authors used quantitative analysis to cluster isolates and to identify potential candidates for future studies relating to the aims of the exercise – composting at low temperatures.

The manuscript is clear written with few editorial matters (references to some analytical methods are not given in some places and versions of software also not noted – this needs to be addressed).  The main issues of concern are as follows:

1.       The definition of ‘psychrotolerant’ is generally accepted as describing bacteria which can grow at temperatures close to 0oC but can grow at temperatures >20oC, often with optima above this (see, for example, Männistö and Puhakka, 2022, doi.org/10.1111/j.1574-6941.2002.tb00961.x).  There are no growth rates at different temperatures and the only mention of impact of temperature was for cellulolytic activity on CMC broths tested at 10-60oC (line 211), although Table 2 reports ‘tolerance tests’ which includes +/- for growth at temperatures in the same temperature range (was this in CMC broth? The Table 2 legend does not say).  Although one could argue that 10oC is ‘close to 0oC’, it is misleading to call the strains psychrotolerant without further evidence that supports this claim.  It is quite common for Bacillus species to grow at the temperatures reported in Table 2. 

2.       Table 2 could be significantly shortened, given the similarity of data between different isolates:  group on the basis of ‘species’, if the data is similar for different isolates then a footnote saying that strains X, Y and Z showed similar trends.

3.       The references in the Introduction quote B. weidmanii (ref 19), which is in the B. cereus group, as ‘psychrotolerant’, with a growth range of 5-43oC.  Close reading of [19] indicates that not all of the strains grew over this temperature range, indicating that the psychrotolerant trait is not a species-wide or species-defining trait (which means that it would not be currently accepted in the rules of nomenclature as defined in IJSEM).  The authors need to be aware of this and be more guarded in accepting published works as currently accurate.

4.       Biochemical tests are reported in Table 1:  are these tests able to distinguish between B. cereus and B. weidmanii?  According to [19], other tests would have been more appropriate – this is clear in the species description.  Why were these particular tests chosen by the authors?  The data in Table 1 only shows that the B. subtilis strains can be differentiated from the others in the ‘top 20’ based on gelatine hydrolysis and that the remainder of the strains cannot be differentiated.  Given that the size/shape of the isolates, and similarity in biochemical tests reported for the non-B. subtilis strains, the Table could be shortened considerably, relegated to supplementary materials and/or summarised in a few words in the text – the level of detail is good but the space required for this Table is redundant.

5.       Fig. 2 shows a phylogram based on sequence similarity of three housekeeping genes.  Many of the strains cluster with B. cereus and are labelled B. weidmanii (as expected), although one in the B. weidmannii cluster is also labelled B. cereus.  Firstly, how significant is the phylogeny based on three housekeeping genes (i.e. is the strain labelled B. cereus actually B. weidmanii?) and, secondly, are many of the strains clonally derived (despite possibly being from the same compost but at different times during composting)?  Given that the strains were selected for growth at 30oC on cellulose, then isolating the same or similar strains over the composting period would not be unexpected.  Clearly, genome sequencing or further physiological characterisation may address this (a future study?).

6.       The measurement of zones of clearing around colonies on plates is fraught with challenges, as the growth rates, diffusion of enzyme, how zone diameters are measured etc may influence the observed outcomes.  This may account for some of the very large error bars seen in Fig. 3.  Many of the methods are well described, except in the detail of what periods of incubation were used when testing traits – some incubation periods were 24-48h (lines 135, 155); 24-72h (line 163) and 20-72h (line 220).  As incubation periods would substantively influence the outcome, can the authors justify why the methods varied so much?  How would this influence the statistical analysis following and the cluster analysis of strains based on cellulolytic and amylolytic activities? 

7.       The authors mention that enzyme activity was also screened at 10oC but there is no data presented.  If the growth curves at different temperatures of some of the strains were included, then data interpretation would have been clearer:  for example, if a strain had a fairly flat relationship between growth rate at temperatures between 15 and 35oC, then it would not be too surprising that enzyme activities would be fairly constant over the optimal temperature range.   

8.       Two-component analysis (Fig. 4) was a useful way of identifying strains with high/low cellulolytic/amylolytic activity at 15 and 20oC, notwithstanding the problems mentioned above in methodology and large error bars on the base data in Fig. 3.

9.       Conclusions:  the most useful observation is that some of the strain of Bacillus made lytic enzymes at a lower temperature than their highest temperature tolerance.  To claim strains are psychrotolerant is misleading and there is no data presented for enzyme production at 10oC.  The isolates were selectively isolated at 30oC from regions that are not particularly cold (given that many publications reporting psychrotolerant strains from cold groundwaters or permanent ice soils) and from soils/compost samples that also were reported to be at about 16-18oC or 29-53oC.  The conclusions need to be in context of the nature of the actual study, what was isolated and the potential of the isolates which are not particularly psychrotolerant.

Overall

There are too many difficulties with the current submission and it would need total restructuring, rewriting and clarification of the focus in context of what was actually found and why.

Author Response

Thanks for giving us the opportunity to submit a revised draft of the manuscript entitled “Cellulolytic and amylolytic bacteria with potential for composting urban solid waste at low temperatures in northern Peru”, paper number: BITED-D-22-00932. I appreciate the time that you and the reviewers have taken to provide valuable comments on this manuscript, which have been considered.

Reviewer 2 Report

The manuscript entitled Isolation of Psychrotolerant Bacteria of Genus Bacillus from Composting Urban Solid Waste and Palm Forest in Northern Peru is interesting for the scientific community. My comments to the authors are listed below.

Keywords: Please give only the relevant keywords for the present manuscript.   

General comments: Please write in italics the name of the bacterial genera and species.

Why did the authors do the PCR reaction only in “a total volume of 10 μL”? Was enough 10 μL of PCR products to perform all the steps for their sequencing (run in the agarose gel, purification, sequencing)?

Please place phenotypic identification of the isolated bacteria before their DNA sequencing and phylogenetic analysis.

Author Response

(The authors gave the same response as above.)

Reviewer 3 Report

In the present manuscript, Vásquez and colleagues isolated psychrotolerant Bacillus from composting and palm forest in Peru. The research scientifically sound and the results are understandable. However, the writing needs some grammatical corrections such as spacing, punctuations, and spell check to improve. It is recommended to ask a native English editor for the emendation. Some examples of these mistakes have been highlighted in the attached file.

-          Please revise the first sentence of the abstract as it is long and a little bit ambiguous.

-          It is highly recommended to explain the novelty of the work in the abstract.

-          Bacterial names should be written in italic form. Please revise any non-italic name in the whole manuscript. E.g. page 2, line 51.

-          It is suggested to provide a table to show the sampling locations explained in subsection 2.1.

-          Please use a unified format to explain amount. In somewhere, g/L is used and in somewhere, % is used. Please revise properly.

-          Why the authors did not screen cellulolytic and amylolytic bacteria at 15 ËšC for the first time and screened them at 30 ËšC? Please indicate this issue in the manuscript.

-          Please add space between number and degree of Celsius for the temperatures used in the manuscript.

-          For the enzyme activity, the unit of each enzyme should be defined. Please add more details about it based on the standard method.

-          Please provide a table for subsection 3.1.

-          In Fig. 3, in some graphs, the error bars are too long that indicates the concentration of the values the average was calculated on is low, and thus that the average value is uncertain. Please reconsider the data.

Author Response

(The authors gave the same response as above.)

Reviewer 4 Report

The study “Isolation of Psychrotolerant Bacteria of Genus Bacillus from Composting Urban Solid Waste and Palm Forest in Northern Peru” describes isolation and characterization of psychrotolerant Bacillus strains with cellulolytic and amylolytic activity to be potentially used for compositing in cold climates. Although this hypothesis was mentioned a few times in the study, the overall impression is that the applied methods are too basic, without bringing too much novelty to the state of the art. For example, cellulolytic and amylolytic activity should be at least expressed in AU of enzymes instead of using only the mass of reducing sugars as the output of the enzymatic reaction. Furthermore, considering the complexity of the composting matrix in terms of available nutrients, potential inhibiting substances and a wide range of other present microorganisms, it would be desirable to have at least some data regarding the concrete composting results presented in the study. Grammar should be checked (especially parts marked in yellow in the attached version), as well as several other aspects mentioned in the comments below. The reviewer’s opinion is that the manuscript would be acceptable for publication in case of adding the concrete results of cellulolytic and amylolytic activity in a more complex composting matrix (mandatory for major revision), otherwise the presented results seem to preliminary.

Microorganisms’ genera and species names should be in italic across the manuscript.

Lines 56-58 – the sentence should be rewritten in a grammatically correct way.

Lines 64-66 – the sentence should be rewritten considering that direct degradation of starch into energy does not happen.

Line 105, 109 – the sentence should not begin with a number, the sentences should be rewritten in a grammatically correct way.

The formula for MC is not correct, please check it. It also misses equation enumeration.

Line 148 – w/v should be moved next to medium composition in %.

It is very unusual to keep pure culture on semi-solid (agarized) medium together with glycerol solution. Please check whether this was the way of stocking the pure cultures, or only biomass of microorganisms in glycerol solution was stocked.

Several interpretations of CI or AI exist in the literature, this one concretely describes the relation between extracellular and intracellular enzymatic activity. Halo zone diameter without biomass diameter divided by the overall halo+growth diameter would better represent only the extracellular activity of the enzymes.

Line 286 - the sentence should not begin with a number.

Although Figure 1 shows grouping of isolates according to their amylolytic and cellulolytic activity, it would be useful to have a table with an overview of all the isolated strains, halo zone diameters and CI/AI, which could be placed in Supplementary, in order to better understand sections 3.2. and 3.3.

Author Response

(The authors gave the same response as above.)

Round 2

Reviewer 1 Report

The authors have provided a revised manuscript which has clarified all of the issues raised in the round 1 review, and they can be congratulated on taking on board the various suggestions to improve the focus of this submission.  I also note that the minor editorial matters raised by other reviews have been addressed but suggest that there may be some errors introduced during rewriting - these will be picked up during the journal’s editorial processes and are minor in nature.  It is recommended that the manuscript is accepted in the present form, with further editorial matters addressed by the journal’s editorial staff.

Reviewer 3 Report

Thanks for improving the manuscript.

Reviewer 4 Report

The authors have responded to the majority of the reviewer's comments.